# Chemical Profile, Bioactivity, and Biosafety Evaluations of Essential Oils and Main Terpenes of Two Plant Species against *Trogoderma granarium*

**Abdulrhman Almadiy [1] and Gomah Nenaah [1,2,\*]**

1 Department of Biology, Faculty of Arts and Sciences, Najran University, Najran 1988, Saudi Arabia
2 Department of Zoology, Faculty of Science, Kafrelsheikh University, Kafr El-Sheikh 33516, Egypt
\* Correspondence: dr_nenaah1972@yahoo.com; Tel.: +966-500-754-240

**Abstract:** In order to search for bio-rational and eco-friendly pest control agents to protect crops from insect infestation, while avoiding the toxic hazards of chemical pesticides, essential oils (EOs) were hydrodistilled from *Juniperus procera* and *Thymus vulgaris* and analyzed using gas chromatography–flame ionization detection (GC–FID), and gas chromatography–mass spectrometry (GC–MS). Eugenol (71.3%), *β*-caryophyllene (11.8%), and *α*-pinene (6.1%) were isolated as the major components of *J. procera* EO, whereas thymol (58.1%), *p*-cymen (10.3%), and carvacrol (8.3%) were the main terpenes in *T. vulgaris* EO. The EOs and terpenes exhibited considerable bioactivity against the khapra beetle using the contact and fumigation bioassays, where *T. vulgaris* EO was superior in bioactivity. Among the terpenes tested, carvacrol and eugenol were superior. Regarding contact toxicity using impregnated filter paper and after 24 h of exposure, the $LC_{50}$ values ranged between 21.4 and 77.0 $\mu L/cm^2$ against larvae and between 16.1 and 69.6 $\mu L/cm^2$ against adults. After 48 h, these values decreased remarkably. Upon fumigation and after 48 h of exposure, the $LC_{50}$ values ranged between 23.0 and 65.3 $\mu L/L$ against larvae, and from 14.2 to 56.4 $\mu L/L$ against adults. The botanicals effectively inhibited the acetylcholinesterase activity of the larvae; however, they were safe for the earthworm *E. fetida* and did not alter the viability of wheat grains. There is a potential for using these botanicals to control *T. granarium.* However, further investigations are needed to confirm the safety of these phytochemicals before use as grain protectants on a commercial scale.

**Keywords:** essential oils; terpenes; *Juniperous procera*; *Thymus vulgaris*; grain protection; khapra beetle; non-target effects

## 1. Introduction

The world's population is increasing steadily, which requires the provision of sufficient food sources to overcome this problem. Cereal crops are the main sources of food for humans worldwide. Therefore, efforts should be made to ensure the sustainability of these prime food resources of humankind. Globally, coleopteran insects are responsible for the greatest overall quantitative and qualitative losses of stored commodities. In temperate and tropical zones, this loss might reach up to 40%. The infestation of grains by harmful insects can also result in the elevated temperature and humidity of grain stores, causing secondary fungal growth and a loss of quality [1]. Grain contamination and impaired germination are also major concerns. The khapra beetle, *Trogoderma granarium* Everts 1898 (Coleoptera: Dermestidae), is a destructive insect and is classified among the 100 worst invasive species of cereal products globally [2,3]. It is a destructive pest that infests not only grain stores (e.g., maize, rice, wheat, and sorghum), but also nuts, walnuts, grams, and pistachios. Infestation extends to dry non-grain commodities, including spices, gums, coconuts, fruits, and other dried protein-rich materials [4]. This beetle is a quarantine pest; hence, strict legislation measures and quarantine protocols are being taken worldwide to prevent its spread. It can survive for a long period of time under extreme conditions

and scarce food resources because of its diapausing late larval stage [4,5]. Accordingly, this pest cannot be easily controlled by conventional insecticides that are effective against other pests of stored products. Moreover, chemical control strategies are facing increasing restrictions because of their environmental problems, hazards against beneficial organisms, high cost, and increasing pest resistance versus chemical pesticides [6–9]. These issues have created a global demand to develop new alternative insecticides, which have to be pest-specific, locally available at a low cost, eco-friendly, non-toxic to non-target species, and less prone to insect resistance [2,3,6]. Recently, research interest has been directed toward plant-based insecticides, since many plants contain several bioactive metabolites with pest control properties [10,11]. Regarding phytochemicals, EOs and/or their main bioactive components, particularly monoterpenes, have exhibited considerable pesticidal properties against stored grain insects and represent novel candidates for pest control strategies with minimal side effects [2,3,6,10–20]. Plant EOs seem appropriate as pest control tools because the majority degrade rapidly in the environment into non-toxic products, hence have low or non-target effects and are safe for mammals and the ecosystem [10,11]. An EO comprises complex mixtures of numerous fractions and accordingly, exerts its insecticidal action through multiple mechanisms; hence, the emergence of resistant strains becomes limited [6]. *Thymus vulgaris* (family: Lamiaceae), commonly known as thyme, is a flowering plant species native to Southern Europe. The plant is indigenous to the Mediterranean basin, Northern Africa, and parts of Asia [21]. It is an essential-oil-bearing plant with a wide range of biological activities, including acaricidal insecticidal, antibacterial, antifungal, and health-promoting activities [22]. Juniper, *Juniperus procera* (family: Cupressaceae) is another widely distributed plant with reported biological activities, such as anticancer, antioxidant, antimicrobial, and insecticidal properties [23]. However, the EO and purified terpenes of *J. procera* and *T. vulgaris* that belong to the Saudi flora have not been evaluated as natural eco-friendly pesticides to control *T. granarium*. The objective of this study was to investigate the chemical composition, insecticidal and acetylcholinesterase inhibitory activities of the EOs hydrodistilled from Saudi *J. procera* and *T. vulgaris* and their terpenes against *T. granarium*, providing insights into their impacts on grain viability and non-target earthworms.

## 2. Materials and Methods

### 2.1. Tested Insect Species

Khapra beetles were reared in a pesticide-free environment for many generations in our laboratory, using sterilized wheat grains (12% moisture) as media. After emergence, 120–150 individual adults were transferred to 1 L capacity glass jars, each provided with 250 g of grains. Rearing jars were covered with muslin cloth and incubated at $28 \pm 2\ °C$, $70 \pm 5\%$ RH and over a 12:12 light/dark photoperiod.

### 2.2. Chemicals

Analytical grade monoterpenes, oxygenated monoterpenes, and sesquiterpenes (Sigma-Aldrich (St. Louis, MO, USA)) were used for peak assignment (see Table 1). A mixture of *n*-alkanes ($C_5$–$C_{30}$) was purchased from Supelco (Bellefonte, PA, USA) and used to calculate the temperature-programmed arithmetic indices of chromatographic peaks.

### 2.3. Tested Plants and EO Extraction

The plant species *J. procera* and *T. vulgaris* were collected from Aseer province, Saudi Arabia in April 2019. The plants were authenticated by our botanists, and voucher specimens (no. Tv 02 and Jp 01) were deposited for further reference. The air-dried samples were powdered using a blender and sieved through a 0.5 mm mesh. Powdered plant samples, weighing 1 kg each, were hydrodistilled in a modified Clevenger apparatus. In each run, 100 g of powder was hydrodistilled using 500 mL of distilled water over a 6 h distillation period. Anhydrous $Na_2SO_4$ was added to remove excess water. Oil yield (% *w/w*) was calculated based on dry weight, and the EOs were kept at $4\ °C$ for bioassays.

**Table 1.** Chemical composition of the plant oils under investigation.

| [a,b] Components | [c] RI Exp. | [d] RI Lit. | Concentration (%) | |
| --- | --- | --- | --- | --- |
| | | | *T. vulgaris* | *J. procera* |
| 2-Hexanal | 856 | 855 | 1.1 | 0.1 |
| Heptanal | 902 | 901 | - | 0.1 |
| *a*-Thujene | 921 | 924 | 1.3 | tr |
| *a*-Pinene | 930 | 932 | 1.1 | 6.1 |
| Camphene | 944 | 946 | 0.4 | – |
| Sabinene | 966 | 969 | tr | 0.1 |
| 1-Octen-3-ol | 976 | 973 | 0.1 | 0.2 |
| β-Myrcene | 986 | 988 | 1.1 | 0.3 |
| *p*-Cymene | 1020 | 1021 | 10.3 | – |
| 1,8-Cineole | 1028 | 1027 | 0.2 | tr |
| (4E,6Z)-allo-Ocimene | 1036 | 1037 | – | 0.4 |
| γ-Terpinene | 1052 | 1054 | 3.4 | – |
| Terpinolene | 1085 | 1086 | 0.2 | 0.3 |
| Linalool | 1096 | 1100 | 1.2 | – |
| Camphor | 1140 | 1141 | 0.2 | 0.2 |
| Pinocarvone | 1158 | 1160 | 0.2 | 0.2 |
| Borneol | 1164 | 1165 | 0.1 | 0.4 |
| Terpinen-4-ol | 1172 | 1174 | 1.1 | 2.7 |
| *a*-Terpineol | 1190 | 1188 | 0.1 | 0.4 |
| Piperitone | 1252 | 1249 | 0.1 | 0.6 |
| Carvone | 1254 | 1252 | 0.3 | - |
| Thymoquinone | 1255 | 1254 | 0.4 | 0.2 |
| Thymol | 1288 | 1289 | 58.1 | – |
| Carvacrol | 1296 | 1298 | 8.3 | – |
| Eugenol | 1355 | 1356 | 0.2 | 71.3 |
| *a*-Copaene | 1371 | 1374 | 0.1 | tr |
| β-Elemene | 1390 | 1389 | – | 0.2 |
| β-Caryophyllene | 1415 | 1417 | 5.2 | 11.8 |
| α-Humulene | 1451 | 1452 | – | 0.4 |
| Acetovanillone | 1462 | 1460 | 0.3 | – |
| Germacrene D | 1480 | 1484 | tr | 0.1 |
| β-Bisabolene | 1508 | 1505 | 0.9 | 0.2 |
| *d*-Cadinene | 1520 | 1522 | 0.2 | – |
| Caryophyllene oxide | 1580 | 1582 | 0.4 | – |
| γ-Cadinol | 1656 | 1655 | 0.5 | – |
| Grouped compounds (%) | - | - | | |
| Monoterpene hydrocarbons | - | - | 20.4% | 7.9% |
| Oxygenated monoterpenes | - | - | 70.3% | 75.7% |
| Sesquiterpene hydrocarbons | - | - | 7.6% | 15.3% |
| Others | - | - | 0.9% | 0% |
| Total | - | - | 99.2% | 98.9% |
| % Yield (mL/100 g dry wt.) | - | - | 1.43 | 0.92 |

[a] EO terpenes are ordered according to their elution in the HP-5MS column. [b] Identification based on: a, comparing RT, RI and MS with authentic standards; and b, comparing mass spectrum of isolated terpenes with data in Wiley NIST 08 MS libraries [24], and Adams' work [25]. [c] The linear retention indices obtained using the HP-5MS column were determined using homologous ($C_5$–$C_{30}$) n-alkane hydrocarbons. [d] The linear retention index [24] or NIST 08 [25], and the literature; tr = trace (<0.05%).

### 2.4. Chemical Analysis of EOs

The aromas of the tested EOs were detected by using an Agilent 6890 N gas chromatograph linked to a flame ionization detector (FID) and an HP-5 capillary column (30 m × 0.32 mm; film thickness: 0.25 μm) (Agilent Technologies, Santa Clara, CA, USA). The operating conditions of the GC–FID were as follows: 1 μL of EO; split mode: split ratio of 50:1; and an injector temperature of 250 °C. Initially, the oven temperature was 40 °C for 3 min, then it was increased to 80 °C at 5°C/min for about 4 min, further increased to

250 °C at 10 °C/min, and then held for 8 min. The temperatures of the injector and detector were set to 250 °C; the carrier gas was helium at a flow rate equal to 1 mL/min. The GC–MS analysis was carried out on a gas chromatograph fused with a silica capillary column (HP-5 MS; 30 m × 0.25 mm i.d.; film thickness: 0.25 μm), linked to a 5975-C mass spectrometer. An amount of 0.1 μL of the EO sample was injected into the capillary column at a split ratio of 1:100. Helium was the carrier gas at a flow rate of 1 mL/min, and the ionization voltage was 70 eV. The temperatures of the ion source and interface were 200 °C and 250 °C, respectively. The mass range was from 45 to 550 AMU, and the oven temperature program was the same as that used in the GC analysis.

### 2.5. Identification of the EO Components

The identification of the EO terpenes was achieved by matching the mass spectra of the individual terpenes with those deposited in the Mass Spectral Library [24], and by comparing the retention indices (LRIs) with those in the literature, using a homologous *n*-alkanes hydrocarbon series ($C_5$–$C_{30}$) from the column [24,25]. Co-injection of the main terpene standards (eugenol, *β*-caryophyllene, *α*-pinene, thymol, *p*-cymene, and carvacrol) dissolved in acetone accessible in our laboratory was also considered. The quantification of individual components was carried out by integrating the FID peak areas and calibrating, and comparing the peak areas with the internal standards without using correction factors. The standard curve for each main terpene was created depending on the concentration and the peak area. The equations of the standard curves and correlation coefficients were determined from a linearity study.

### 2.6. Isolation and Purification of Main Terpenes

A total of 15 mL of each of the tested EOs was chromatographed on a silica gel column (kieslgel 60; 230–400 mesh, Merck, Darmstadt, Germany). For elution, we used a mobile phase that consisted of *n*-hexane/acetone, starting with fractions of 100 mL *n*-hexane (10×), 2% acetone/*n*-hexane (20×), 5% acetone/*n*-hexane (10×), 10% acetone/*n*-hexane (5×), and finally, an acetone solvent system (5×). The developed fractions were pooled in 3 main fractions based on the TLC profiles. Fractions with insecticidal bioactivity, and similar TLC profiles were gathered and purified using pre-coated thin-layer silica chromatographic plates (PTLCs). For *J. procera*, the first fraction (fractions 9–14, 3.9 g) was purified using a silica gel column eluted with a solution of 2% acetone/*n*-hexane to give 2.3 g of eugenol. The second developed fraction (fractions 20–26, 1.1 g) was developed using 10% acetone/*n*-hexane as an eluent to produce 0.45 g of *β*-caryophyllene. The third fraction (fractions 31–35, 0.65 g) was eluted with 2% *n*-hexane/acetone/chloroform (1:0.1:0.1) to give 0.26 g of *α*-pinene. The purification of the bioactive fractions of *T. vulgaris* EO yielded thymol (2.7 g), *p*-cymene (0.86 g), and carvacrol (0.82 g). The structures of the terpenes were established using spectroscopy, including [1]H and [13]C-NMR (Bruker AMX500 (500 MHz ([1]H)) instrument), using $CDCl_3$ or DMSO-d6 as a solvent, and TMS as the internal standard, and HR-MS.

### 2.7. Contact Toxicity

The contact bioactivities of the oils and main terpenes were investigated using impregnated filter papers [2]. Test solutions of each EO or fraction were made by dissolving 3.18, 1.59, 0.795, 0.398, 0.2, and 0.1 mL of each phytochemical in acetone (5 mL). Each concentration was applied uniformly to a Whatman No. 1 filter paper disc (4.5 cm diameter, 31.8 $cm^2$) to obtain final concentrations ranging from 50.0 to 1.56 μL/$cm^2$. The papers were dried at ambient temperature and then confined separately to 4.5 cm diameter Petri dishes. Twenty (unsexed, 5-days old) adults or second-stage larvae were separately confined to the treated Petri dishes. Control groups (filter papers treated with acetone) were established. The treated and control groups were incubated under the rearing conditions previously described. The insects were returned to sterilized Petri dishes provided with food 24 h post-exposure and incubated under rearing conditions. For each experiment and

control, 6 replicates were performed. Mortality was counted 24 and 48 h post-treatment. Insects were considered dead if no appendage movement was observed after prodding with a brush.

### 2.8. Fumigation

The bioactivities of the oils and pure compounds were evaluated via fumigation [3]. Filter papers, 7 cm in diameter, were soaked in 25 µL of 6 serial dilutions of each phytochemical to obtain equivalent fumigant concentrations equal to 91.7, 45.85, 22.9, 11.5, 5.7, and 2.85 µL/L of air or acetone only (control). After solvent evaporation, each treated piece of paper was fixed to the underside of a screw cap of a 50 mL volume glass vial. The insects were introduced to the vials in groups of 20 adults (5 days old, unsexed) or 2nd instar larvae. The vials were covered with fine steel gauze fixed with tape. The insects were returned to sterilized vials provided with food 24 h after exposure and incubated under rearing conditions. For each experiment, 6 replicates were performed with the control groups, and mortality was counted 24 and 48 h post-treatment.

### 2.9. Acetylcholinesterase (AChE) Inhibition

About 0.6 g of *T. granarium* larvae was obtained when the insect was homogenized in 15 mL of 50 mM phosphate buffer (pH 7.5). The inhibition of AChE activity was monitored calorimetrically using the supernatant as a source for the enzyme [26], and acetylthiocholine iodide (25 µL, 15 mM) as a substrate. The EO terpenes were dissolved in acetone and Triton-X 100 (0.01%), and then they were assayed at 2.5~100 mM. The experiments were replicated 3 times and the specific activity of AChE ($\Delta$OD/mg protein/min) was monitored at 412 nm.

### 2.10. Toxicity against a Non-Target Terrestrial Organism

The acute toxic effects of the EO materials were assessed against the earthworm *E. fetida*, following the OECD guidelines (Organization for Economic Co-operation and Development) [27]. The earthworm was cultured on artificial soil as described before [28]. Test materials were mixed with dry soil at 50, 100, and 200 mg kg$^{-1}$. The chemical pesticide, $\alpha$-cypermethrin, at 10 and 20 mg kg$^{-1}$, was included as a positive control, while de-ionized water was a negative control. In 1 L glass pots, 10 adult earthworms were introduced into the soil (treated and control), and 4 replicates were carried out for each trial ($n = 40$). The pots were covered with gauze and incubated at $20 \pm 1$ °C, and 80–85% RH and over a 16:8 light/dark photoperiod, and mortality was counted 5 and 10 days post-treatment.

### 2.11. Phytotoxicity Study

The phytotoxic effects of EOs and bioactive terpenes on the germination, and subsequent radicle and shoot growth of the wheat plant *Triticum aestivum* L. were measured. Before treatments, seeds were soaked in (15%) sodium hypochlorite solution for 5 min, and then rinsed in de-ionized H$_2$O to avoid any possible growth inhibition related to fungal toxins. The seeds were sown in a sterilized 9 cm Petri plate, containing 5 layers of Whatman filter paper. Two mL of each EO or terpene was poured on the filter paper at 50, 100, and 200 µL/mL in treated groups or 2 mL of methanol for the control. After the evaporation of the solvent, 12 mature healthy seeds of wheat plants were introduced to each plate and allowed to germinate at $22 \pm 2$ °C over a natural photoperiod. Every day, the growth plates were examined for seed germination, with germination being determined when root protrusion became evident. Each concentration was replicated 4 times with the controls, and impacts on the germination, root, and shoot lengths were recorded on the fourth day.

### 2.12. Data Analysis

Abbott's formula [29] was adopted to correct the mortality (% means $\pm$ S.E.) relative to that of the control. A probit analysis was used to analyze the dose–mortality response [30]. LC$_{50}$ and LC$_{95}$ values were estimated based on the 24 and 48 h exposure periods. Con-

centrations that caused 50% AChE inhibition ($IC_{50}$) were estimated by adopting a linear regression analysis. The phytotoxicity data were interpreted as means (±S.E.), and significances of mean differences between the treated and control groups were compared using ANOVA ($\alpha$ = 0.05), followed by individual pairwise comparisons using Tukey's-b honest significant differences test. A data analysis was performed using the Statistical Package Social Science (SPSS) software, version 23.

## 3. Results

### 3.1. Aroma Profile of EOs

The EOs hydrodistilled from *J. procera* and *T. vulgaris* were yielded at 0.92 and 1.43%, respectively (Table 1). *J. procera* EO mainly contained 2-methoxy-4-prop-2-enylphenol (eugenol, 71.3%), (1R,4E,9S)-4,11,11-trimethyl-8-methylidenebicyclo[7.2.0]undec-4-ene (ß-caryophyllene, 11.8%), and 2,6,6-trimethylbicyclo[3.1.1]hept-2-ene (*a*-pinene, 6.1%). Meanwhile, 5-methyl-2-(propane-2-yl) phenol (thymol, 58.1%), 1-methyl-4-(propane-2-yl) benzene (*p*-cymene, 10.3%), and 2-methyl-5-(propane-2-yl) phenol (carvacrol, 8.3%) were the main terpenes of *T. vulgaris* EO [Figure 1]. The bioactive terpenes mainly belong to the following three groups: monoterpenes, oxygenated monoterpenes, and sesquiterpenes, where oxygenated monoterpenes were abundant in both EOs. The spectroscopic data of the terpenes were confirmed using physical and spectroscopic tools, and they can be summarized as follows:

(**A**)

*P*-Cymene Carvacrol Thymol

(**B**)

Eugenol β-Caryophyllene $\alpha$-Pinene

**Figure 1.** Bioactive terpenes isolated from (**A**) *T. vulgaris* and (**B**) *J. procera* EOs.

### 3.1.1. Eugenol

Yellowish oil. [1]H NMR (600 MHz, CDCl$_3$): δ 6.82 (H-6, d, *J* = 8.4 Hz), 6.65–6.69 (2H, m, H3, H-5), 3.31 (H2-1′, d, *J* = 6.9 Hz), 5.92 (H-2′, m), 5.04–5.06 (H2-3′, m), 5.46 (s, 1-OH), 3.84 (2-OCH$_3$, s); [13]C NMR (150 MHz, CDCl$_3$): δ 142.86 (C-1), 147.30 (C-2), 111.09 (C-3), 131.81 (C-4), 121.19 (C-5), 113.90 (C-6), 38.98 (C-1′), 136.88 (C-2′), 115.41 (C-3′), 55.74 (2-OCH$_3$) [31].

### 3.1.2. β-Caryophyllene

Colorless oil. [1]H NMR (600 MHz, CDCl$_3$) δ: 1.67 (H2-1), 1.43, 1.53 (H2-2), 1.91, 2.08 (H2-3), 5.32 (H-5,dd, 4.1, 9.7 Hz), 2.02, 2.37 (H2-6), 2.03, 2.24 (H2-7), 2.31 (H-9); 1.61, 1.67 (H2-10), 0.97 (H3-12), 0.95 (H3-13), 1.61 (H3-14), 4.83, 4.94 (H2-15); [13]C NMR (150 MHz, CDCl$_3$) δ: 53.6 (C-1), 29.4 (C-2), 39.8 (C-3), 135.4 (C-4), 123.8 (C-5), 28.7 (C-6), 34.6 (C-7), 154.8 (C-8), 48.4 (C-9), 40.2 (C-10), 33.1 (C-11), 22.6 (C-12), 30.2 (C-13), 16.2 (C-14), 111.7 (C-15) [32].

### 3.1.3. *α-Pinene*

Colorless oil. $^1$H NMR (CDCl$_3$, 300 MHz): δ 1.92 (m, 1H), δ 1.4 (s, 3H), δ 1.6 (s, 3H), δ 1.8 (s, 3H), δ 1.9 (m, 2H), δ 2.3 (m, 1H), δ 2.4 (m, 1H), δ 4.2 (s, 1H), δ 5.6 (t, 1H), 20.67, 22.35, 22.47, 26.36, 32.62, 36.08, 68.03, 94.15, 123.87, 134.38; $^{13}$C NMR (125 MHz, CHCl$_3$): δ 46.99 (C-1), 144.6 (C-2), 116.0 (C-3), δ 31.3 (C-4), 40.69 (C-5), 37.97(C-6), 31.5 (C-7), δ 26.3 (C-8), 20.8 (C-9), 23.01 (C-10) [33,34].

### 3.1.4. *Carvacrol*

Deep-orange oil. $^1$H NMR (600 MHz, CHCl$_3$-*d*): δ (ppm) = 1.39 (6H, m, H-8, H-9), 2.43 (3H, s, H-10), 2.98 (1H, m, H-7), 5.88 (1H, br s, OH), 6.84 (1H, s, H-6), 6.93 (1H, d, *J* = 7.3 Hz, H-4), 7.25 (1H, d, *J* = 7.3 Hz, H-3); $^{13}$C NMR (125 MHz, CHCl$_3$-*d*): δ 15.51 q (C-10), 24.03 q (C-8, C-9), 33.77 d (C-7),113.35 d (C-6), 119.12 d (C-4), 121.44 s (C-2), 131.02 d (C-3), 148.47 s (C-5), 153.46 s (C-1) [35].

### 3.1.5. *Thymol*

Colorless oil. $^1$H NMR (400 MHz, CHCl$_3$-*d*): δ (ppm) = 5.40 H(1), 6.19 H(2), 7.08 H(3), 3.38 H(4), 1.05 H(5–7), 1.45 H(8–10), 0.42 H(11–13), 9.99 H(14); $^{13}$C NMR (125 MHz, CHCl$_3$): δ 150.2 (C-1), 116.9 (C-2), 138.4 (C-3), 123.6 (C-4), 126.3 (C-5), 131.7 (C-6), 25.5 (C-7), 26.1 (C-8), 23.6 (C-9), 18.7 (C-10) [36].

### 3.1.6. *p-Cymene*

Colorless oil. $^1$H NMR (600 MHz, CHCl$_3$-*d*): δ (ppm) = 5.40 (H1), 7.11 H(2), 7.24 H(3), 7.24 H(5), 7.162 H(6), 2.252 H(7), 1.381 H(9), 1.381 H(10); $^{13}$C NMR (125 MHz, CHCl$_3$): δ 140.86 (C-1), 128.72 (C-2), 124.30 (C-3), 149.27 (C-4), 124.30 (C-5), 128.77 (C-6), 21.31 (C-7), 70.89 (C-8), 31.78 (C-9), 31.64 (C-10) [37].

### *3.2. Insecticidal Activity*

The contact bioactivity data (Tables 2 and 3) show that *T. vulgaris* EO was superior as an insecticide (LC50 (larvae) = 15.1 μL/cm$^2$, $\chi^2$ = 2.2; *df* = 7), and (LC$_{50}$ (adult) = 11.5 μL/cm$^2$, $\chi^2$ = 1.38; *df* = 7). LC$_{50}$ values of *J. procera* EO were (24.5 μL/cm$^2$, $\chi^2$ = 2.11; *df* = 7, larvae), and (19.5 μL/cm$^2$, $\chi^2$ = 1.15; *df* =7, adult). Regarding the bioactive terpenes, carvacrol was superior (LC$_{50}$ larvae = 21.2 μL/cm$^2$, $\chi^2$ = 1.70; *df* = 7), and (LC$_{50}$ adult = 14.6 μL/cm$^2$, $\chi^2$ = 0.55; *df* = 7) after 48 h of exposure. Eugenol was an effective insecticide (LC$_{50}$ larvae = 30.6 μL/cm$^2$, $\chi^2$ = 2.21; *df* = 7), and (LC$_{50}$ adult = 22.4 μL/cm$^2$, $\chi^2$ = 1.24; *df* = 7). Using fumigation, the larval LC$_{50}$ ranged from 23.0 to 65.3 μL/L after 48 h (Table 4), while the adults were highly susceptible, with LC$_{50}$ values ranging from 14.2 to 56.4 μL/L (Table 5). In this case, the EO of *T. vulgaris* was the most potent (LC$_{50}$ larvae = 23.0 μL/L, $\chi^2$ = 1.04; *df* = 7, and LC$_{50}$ adult = 14.2 μL/L, $\chi^2$ = 1.66; *df* = 7), followed by carvacrol (larval LC$_{50}$ = 29.5 μL/L, $\chi^2$ = 2.11; *df* = 5, and adult LC$_{50}$ = 22.4 μL/L, $\chi^2$ = 2.02; *df* = 7), *J. procera* oil (larval LC$_{50}$ = 31.3 μL/L, $\chi^2$ = 1.25; *df* = 7, and adult LC$_{50}$ = 21.8 μL/L, $\chi^2$ = 1.68; *df* = 7), and eugenol (larval LC$_{50}$ = 40.6 μL/L, $\chi^2$ = 2.17; *df* = 7, and adult LC$_{50}$ = 30.2 μL/L, $\chi^2$ = 1.43; *df* = 7).

**Table 2.** Contact larvicidal activity ($\mu$L/cm$^2$) of the tested plant oils and their terpenes against *T. granarium* using the filter paper bioassay.

| Plant Oil (Fractions) | | 24 h | | | | 48 h | | | | |
|---|---|---|---|---|---|---|---|---|---|---|
| | | LC$_{50}$ (95% fl) | LC$_{95}$ (95% fl) | Slope ($\pm$ S.E.) | Chi$^2$ (df = 7) | LC$_{50}$ (95% fl) | LC$_{95}$ (95% fl) | Slope ($\pm$ S.E.) | Chi$^2$ (df = 7) | p |
| *T. vulgaris* | EO | 21.4 (17.1–27.5) | 53.3 (48.3–64.9) | 2.0 $\pm$ 0.26 | 0.47 | 15.1 (12.6–21.8) | 39.3 (33.5–47.4) | 1.8 $\pm$ 0.21 | 2.20 | 0.345 |
| | Carvacrol | 26.1 (22.3–32.5) | 58.2 (52.2–70.8) | 2.1 $\pm$ 0.22 | 2.03 | 21.2 (18.1–33.5) | 48.7 (41.0–60.4) | 2.0 $\pm$ 0.20 | 1.70 | 0.408 |
| | Thymol | 55.7 (46.8–64.1) | 127.9 (118.2–143.1) | 2.2 $\pm$ 0.21 | 2.44 | 46.4 (40.3–55.8) | 109.1 (97.0–123.3) | 2.4 $\pm$ 0.18 | 2.06 | 0.349 |
| | *P*-Cymene | 81.7 (75.6–93.3) | 188.5 (175.7–204.1) | 3.0 $\pm$ 0.38 | 3.26 | 70.6 (65.6–82.5) | 164.8 (151.3–181.5) | 2.7 $\pm$ 0.28 | 3.08 | 0.538 |
| *J. procera* | EO | 33.7 (28.1–39.0) | 70.3 (60.0–82.7) | 2.5 $\pm$ 0.30 | 1.34 | 24.5 (20.7–30.2) | 61.2 (53.3–71.1) | 2.6 $\pm$ 0.28 | 2.11 | 0.302 |
| | Eugenol | 37.5 (33.9–42.7) | 83.5 (78.2–94.4) | 3.4 $\pm$ 0.40 | 2.36 | 30.6 (25.3–36.7) | 70.0 (65.8–82.2) | 2.1 $\pm$ 0.24 | 2.21 | 0651 |
| | $\alpha$-Pinene | 60.6 (53.9–68.2) | 147.2 (134.0–163.1) | 2.4 $\pm$ 0.37 | 3.22 | 54.1 (47.6–62.3) | 128.3 (116.7–152.9) | 2.8 $\pm$ 0.22 | 3.13 | 0.493 |
| | $\beta$-Caryophyllene | 77.0 (69.3–91.9) | 162.4 (146.5–177.3) | 2.2 $\pm$ 0.35 | 1.81 | 62.4 (53.7–73.6) | 133.3 (121.4–150.1) | 2.7 $\pm$ 0.17 | 3.62 | 0.611 |

LC$_{50}$ and LC$_{95}$ are considered to be significantly different when the 95% fiducial limits (fl) do not overlap. Values are the mean of 5 replicates, each set up with 20 individuals (*n* = 100).

**Table 3.** Contact adulticidal activity ($\mu$L/cm$^2$) of the tested oils and their terpenes against *T. granarium* using the filter paper bioassay.

| Plant Oil (Fractions) | | 24 h | | | | 48 h | | | | |
|---|---|---|---|---|---|---|---|---|---|---|
| | | LC$_{50}$ (95% fl) | LC$_{95}$ (95% fl) | Slope ($\pm$ S.E.) | Chi$^2$ (df = 7) | LC$_{50}$ (95% fl) | LC$_{95}$ (95% fl) | Slope ($\pm$ S.E.) | Chi$^2$ (df = 7) | p |
| *T. vulgaris* | EO | 16.1 (13.0–21.8) | 42.4 (34.3–48.2) | 2.1 $\pm$ 0.42 | 1.32 | 11.5 (9.1–14.2) | 30.1 (23.1–35.5) | 2.1 $\pm$ 0.32 | 1.38 | 0.399 |
| | Carvacrol | 18.8 (16.0–22.6) | 43.4 (39.1–51.5) | 1.6 $\pm$ 0.48 | 0.76 | 14.6 (12.2–18.3) | 34.8 (30.3–42.2) | 1.6 $\pm$ 0.30 | 0.55 | 0.502 |
| | Thymol | 42.5 (35.3–50.1) | 95.2 (82.2–114.9) | 2.4 $\pm$ 0.51 | 1.84 | 37.3 (30.8–42.5) | 80.4 (73.2–93.5) | 2.5 $\pm$ 0.48 | 1.42 | 0.411 |
| | *P*-Cymene | 70.6 (62.8–84.1) | 161.4 (149.6–178.0) | 3.7 $\pm$ 0.24 | 3.15 | 58.3 (49.6–71.4) | 135.5 (124.7–150.9) | 2.5 $\pm$ 0.36 | 1.15 | 0.713 |
| *J. procera* | EO | 25.5 (19.7–32.6) | 64.9 (56.4–74.3) | 2.3 $\pm$ 0.47 | 1.22 | 19.5 (17.3–23.2) | 49.0 (43.7–58.1) | 2.6 $\pm$ 0.62 | 1.15 | 0.420 |
| | Eugenol | 28.6 (24.3–33.1) | 67.1 (61.4–77.8) | 2.1 $\pm$ 0.26 | 1.12 | 22.4 (19.6–26.7) | 57.7 (52.3–68.2) | 2.0 $\pm$ 0.21 | 1.24 | 0.581 |
| | $\alpha$-Pinene | 50.4 (45.3–63.5) | 121.8 (113.0–134.4) | 2.4 $\pm$ 0.18 | 2.87 | 44.9 (38.1–54.3) | 110.6 (101.3–121.6) | 2.2 $\pm$ 0.22 | 2.35 | 0.403 |
| | $\beta$-Caryophyllene | 69.6 (63.7–77.8) | 151.1 (136.1–170.1) | 2.9 $\pm$ 0.37 | 3.15 | 51.6 (45.3–63.8) | 122.5 (113.9–136.7) | 3.1 $\pm$ 0.28 | 3.11 | 0.719 |

LC$_{50}$ and LC$_{95}$ are considered to be significantly different when the 95% fiducial limits (fl) do not overlap. Values are the mean of 5 replicates, each set up with 20 individuals (*n* = 100).

**Table 4.** Fumigant larvicidal activity (µL/L) of the tested plant oils and their major terpenes against *T. granarium*.

| Plant Oil (Fractions) | | 24 h | | | | 48 h | | | | |
|---|---|---|---|---|---|---|---|---|---|---|
| | | LC$_{50}$ (95% fl) | LC$_{95}$ (95% fl) | Slope (± S.E.) | Chi$^2$ (df = 7) | LC$_{50}$ (95% fl) | LC$_{95}$ (95% fl) | Slope (± S.E.) | Chi$^2$ (df = 7) | p |
| *T. vulgaris* | EO | 30.4 (27.1–34.5) | 58.3 (49.3–66.9) | 2.0 ± 0.26 | 0.47 | 23.0 (18.6–27.8) | 47.3 (41.5–56.4) | 1.04 | 1.04 | 0.567 |
| | Carvacrol | 37.7 (31.1–45.0) | 80.3 (70.0–89.7) | 2.5 ± 0.30 | 1.34 | 29.5 (24.7–34.2) | 69.2 (64.3–77.1) | 2.11 | 2.11 | 0.409 |
| | Thymol | 58.2 (50.6–67.9) | 132.1 (124.1–143.5) | 2.4 ± 0.24 | 2.60 | 47.6 (39.4–55.5) | 99.3 (88.1–112.3) | 2.43 | 2.43 | 0.456 |
| | *P*-Cymene | 73.6 (65.7–82.8) | 162.1 (146.1–183.1) | 2.9 ± 0.37 | 3.15 | 63.6 (56.3–71.8) | 141.5 (130.9–162.7) | 3.11 | 3.11 | 0.381 |
| *J. procera* | EO | 39.5 (34.4–46.2) | 82.6 (76.8–91.2) | 2.2 ± 0.22 | 1.65 | 31.3 (27.3–35.7) | 63.1 (57.6–71.8) | 1.25 | 1.25 | 0.623 |
| | Eugenol | 49.2 (42.6–52.9) | 110.1 (97.1–127.5) | 2.4 ± 0.24 | 2.60 | 40.6 (36.4–47.5) | 88.7 (80.1–100.5) | 2.17 | 2.17 | 0.461 |
| | α-Pinene | 61.0 (55.2–66.1) | 150.8 (138.6–174.0) | 2.4 ± 0.22 | 2.63 | 51.8 (42.3–57.4) | 134.1 (122.6–144.1) | 3.73 | 3.73 | 0.388 |
| | β-Caryophyllene | 77.0 (70.3–84.5) | 213.4 (194.7–236.6) | 2.2 ± 0.20 | 5.66 | 65.3 (60.2–76.6) | 172.3 (158.5–187.3) | 2.9 ± 0.41 | 3.74 | 0.644 |

LC$_{50}$ and LC$_{95}$ are considered to be significantly different when the 95% fiducial limits (fl) do not overlap. Values are the mean of 5 replicates, each set up with 20 individuals (*n* = 100).

**Table 5.** Fumigant adulticidal activity (µL/L) of the tested plant oils and their major terpenes against *T. granarium*.

| Plant Oil (Fractions) | | 24 h | | | | 48 h | | | | |
|---|---|---|---|---|---|---|---|---|---|---|
| | | LC$_{50}$ (95% fl) | LC$_{95}$ (95% fl) | Slope (± S.E.) | Chi$^2$ (df = 7) | LC$_{50}$ (95% fl) | LC$_{95}$ (95% fl) | Slope (± S.E.) | Chi$^2$ (df = 7) | p |
| *T. vulgaris* | EO | 21.6 (17.1–26.5) | 50.3 (43.3–60.9) | 2.0 ± 0.26 | 0.47 | 14.2 (11.6–20.8) | 40.3 (34.5–49.4) | 1.8 ± 0.21 | 1.66 | 0.643 |
| | Carvacrol | 29.2 (28.1–38.0) | 73.1 (65.0–83.7) | 2.5 ± 0.30 | 1.34 | 22.4 (17.7–29.2) | 59.2 (51.3–74.1) | 2.6 ± 0.28 | 2.02 | 0.498 |
| | Thymol | 50.8 (46.6–62.9) | 118.1 (119.1–147.5) | 2.4 ± 0.24 | 2.60 | 2.6 (31.4–43.5) | 90.3 (84.1–104.3) | 2.6 ± 0.24 | 2.43 | 0.823 |
| | *P*-Cymene | 65.6 (58.7–74.8) | 150.1 (139.1–169.1) | 2.9 ± 0.37 | 3.15 | 54.6 (47.3–60.8) | 127.5 (118.9–143.7) | 3.1 ± 0.28 | 3.11 | 0.754 |
| *J. procera* | EO | 26.5 (23.2–32.5) | 70.4 (64.4–74.4) | 1.8 ± 0.20 | 0.35 | 21.8 (22.2–31.2) | 54.3 (48.2–63.3) | 2.1 ± 0.19 | 1.68 | 0.532 |
| | Eugenol | 41.4 (36.5–50.6) | 91.6 (84.6–108.5) | 2.2 ± 0.30 | 1.86 | 30.2 (26.5–37.0) | 80.1 (72.5–93.2) | 2.0 ± 0.30 | 1.43 | 0.672 |
| | α–Pinene | 55.6 (47.2–64.7) | 139.3 (128.0–151.4) | 3.0 ± 0.31 | 3.88 | 42.5 (36.0–51.4) | 118.5 (109.4–130.5) | 2.0 ± 0.25 | 2.34 | 0.896 |
| | β–Caryophyllene | 70.1 (65.8–78.9) | 182.5 (171.6–201.0) | 2.0 ± 0.26 | 3.42 | 56.4 (50.2–68.6) | 137.6 (126.7–153.3) | 3.1 ± 0.33 | 2.78 | 0.322 |

LC$_{50}$ and LC$_{95}$ are considered to be significantly different when the 95% fiducial limits (fl) do not overlap. Values are the mean of 5 replicates, each set up with 20 individuals (*n* = 100).

### 3.3. AChE Inhibition

The data on AChE inhibition caused by the EOs and their bioactive terpenes, in terms of IC$_{50}$ values (Table 6), revealed that the EOs and terpenes altered the larval AChE bioactivity of *T. granarium*. The EOs of *T. vulgaris* (IC$_{50}$ = 18.23 mg/L, $\chi^2$ = 2.22, *df* = 8; *p* = 0.733) and carvacrol (IC$_{50}$ = 19.08 mg/L, $\chi^2$ = 1.21, *df* = 8; *p* = 614) were found to be potent AChE inhibitors, followed by *J. procera* EO (IC$_{50}$ = 23.22 mg/L, $\chi^2$ = 2.63, *df* = 8; *p* = 526), and eugenol (IC$_{50}$ equal to 26.16 mg/L, $\chi^2$ = 2.33, *df* = 8; *p* = 565). The remaining terpenes demonstrated moderate to weak larval AChE inhibition, with IC$_{50}$ values ranging from 30.14 to 59.78 mg/L.

**Table 6.** Acetylcholinesterase (AChE) inhibition in *T. granarium* larvae by the tested oils and their terpenes.

| Plant Oil (Fractions) | | [a] IC$_{50}$ (mg/L) | (95% Fiducial Limits) | Slope (±S.E.) | *Chi*$^2$ (*df* = 8) | *p* |
|---|---|---|---|---|---|---|
| *T. vulgaris* | EO | 18.23 | (15.08–22.44) | 2.06 ± 0.33 | 2.22 | 0.733 |
| | Carvacrol | 19.08 | (17.21–23.50) | 1.08 ± 0.19 | 1.21 | 0.614 |
| | Thymol | 30.14 | (24.08–37.44) | 2.26 ± 0.33 | 3.35 | 0.723 |
| | *P*-Cymene | 42.00 | (37.08–53.21) | 2.26 ± 0.33 | 3.22 | 0.523 |
| *J. procera* | EO | 23.22 | (20.33–27.91) | 1.16 ± 0.23 | 2.63 | 0.526 |
| | Eugenol | 26.16 | (23.84–31.12) | 1.12 ± 0.27 | 2.33 | 0.565 |
| | *a*-Pinene | 34.93 | (31.14–41.82) | 1.18 ± 0.19 | 3.25 | 0.245 |
| | *β*-Caryophyllene | 59.78 | (54.21–72.50) | 2.08 ± 0.19 | 3.21 | 0.544 |
| | Methomyl | $2.24 \times 10^{-3}$ | ($1.66 \times 10^{-3}$–$3.30 \times 10^{-3}$) | 2.02 ± 0.12 | 2.33 | 0.423 |

[a] The concentration that causes 50% enzyme inhibition.

### 3.4. Toxicity against Earthworm

The toxicity of the EO products was evaluated against the earthworm, *E. fetida*, as a terrestrial non-target species. The botanicals were found to be safe for *E. fetida*, as no mortality or toxicity symptoms were observed on the tested animals following the application of phytochemicals, even at 200 mg kg$^{-1}$ (Table 7).

**Table 7.** * Acute toxicity of plant oils and their main terpenes against the earthworm, *Eisenia fetida*.

| Plant Material | | % Mortality (Mean ± S.E.) at Different Concentrations (mg kg$^{-1}$) | | | | | |
|---|---|---|---|---|---|---|---|
| | | 50 | | 100 | | 200 | |
| | | 5 Days | 10 Days | 5 Days | 10 Days | 5 Days | 10 Days |
| *T. vulgaris* | EO | 0.0 ± 0.0 [c] | 0.0 ± 0.0 [c] | 0.0 ± 0.0 [c] | 0.0 ± 0.0 [c] | 0.0 ± 0.0 [c] | 0.0 ± 0.0 [c] |
| | Carvacrol | 0.0 ± 0.0 [c] | 0.0 ± 0.0 [c] | 0.0 ± 0.0 [c] | 0.0 ± 0.0 [c] | 0.0 ± 0.0 [c] | 0.0 ± 0.0 [c] |
| | Thymol | 0.0 ± 0.0 [c] | 0.0 ± 0.0 [c] | 0.0 ± 0.0 [c] | 0.0 ± 0.0 [c] | 0.0 ± 0.0 [c] | 0.0 ± 0.0 [c] |
| | *P*-Cymene | 0.0 ± 0.0 [c] | 0.0 ± 0.0 [c] | 0.0 ± 0.0 [c] | 0.0 ± 0.0 [c] | 0.0 ± 0.0 [c] | 0.0 ± 0.0 [c] |
| *J. procera* | EO | 0.0 ± 0.0 [c] | 0.0 ± 0.0 [c] | 0.0 ± 0.0 [c] | 0.0 ± 0.0 [c] | 0.0 ± 0.0 [c] | 0.0 ± 0.0 [c] |
| | Eugenol | 0.0 ± 0.0 [c] | 0.0 ± 0.0 [c] | 0.0 ± 0.0 [c] | 0.0 ± 0.0 [c] | 0.0 ± 0.0 [c] | 0.0 ± 0.0 [c] |
| | *a*-Pinene | 0.0 ± 0.0 [c] | 0.0 ± 0.0 [c] | 0.0 ± 0.0 [c] | 0.0 ± 0.0 [c] | 0.0 ± 0.0 [c] | 0.0 ± 0.0 [c] |
| | *β*-Caryophyllene | 0.0 ± 0.0 [c] | 0.0 ± 0.0 [c] | 0.0 ± 0.0 [c] | 0.0 ± 0.0 [c] | 0.0 ± 0.0 [c] | 0.0 ± 0.0 [c] |
| Negative control (water) | | 0.0 ± 0.0 [c] | 0.0 ± 0.0 [c] | 0.0 ± 0.0 [c] | 0.0 ± 0.0 [c] | 0.0 ± 0.0 [c] | 0.0 ± 0.0 [c] |
| *α*-Cypermethrin 10.0 mg kg$^{-1}$ | | 30.8 ± 2.4 [b] | 36.2 ± 2.3 [b] | 54.9± 1.9 [b] | 68.3 ± 2.6 [b] | 82.4 ± 2.2 [b] | 90.1 ± 1.6 [a] |
| *α*-Cypermethrin 20.0 mg kg$^{-1}$ | | 64.2 ± 3.3 [a] | 70.7 ± 2.1 [a] | 88.1 ± 2.0 [a] | 100.0 ± 0.0 [a] | 100.0 ± 0.0 [a] | 100.0 ± 0.0 [a] |
| * *F*-value | | 1219.1 | 902.1 | 1973.9 | 2001.8 | 5765.9 | 8810.7 |
| *p*-value | | 0.004 | 0.006 | 0.002 | 0.007 | 0.004 | 0.003 |

* Each value represents the mean of 4 replicates, each with 10 individuals (*n* = 40). In the same column, means followed by the same letter(s) are not significantly different (*p* ≤ 0.05) (Tukey's HSD test). All *F* values were significant (*p* ≤ 0.001).

### 3.5. Phytotoxicity

The phytotoxicity data (Table 8) revealed that the tested EOs and their major terpenes were not phytotoxic to the wheat plant, as the (%) germination, radical, and shoot growth of the wheat grains were not significantly affected after the treatment of grains with the

tested botanicals at 50.0 and 100 μg/mL. At a concentration of 200 μg/mL, the percentage germination and the shoot growth were slightly affected, particularly by the EO of *T. vulgaris* and carvacrol.

**Table 8.** * Phytotoxic activity of the tested plant oils and their fractions against wheat plants.

| Test Material | | Concentration (μg/mL) | Germination (%) | RL | SL |
|---|---|---|---|---|---|
| *T. vulgaris* | EO | 50 | 90.7 ± 1.3 [a] | 8.98 ± 0.38 [a] | 3.35 ± 0.13 [a] |
| | | 100 | 83.3 ± 1.4 [ab] | 8.35 ± 0.20 [a] | 3.24 ± 0.12 [a] |
| | | 200 | 74.4 ± 1.3[b] | 8.02 ± 0.19 [a] | 2.48 ± 0.12 [ab] |
| | Carvacrol | 50 | 89.9 ± 2.3 [a] | 9.04 ± 0.22 [a] | 3.30 ± 0.12 [a] |
| | | 100 | 80.8 ± 1.2 [ab] | 8.85 ± 0.21 [a] | 2.91 ± 0.14 [a] |
| | | 200 | 71.3 ± 1.2[b] | 8.06 ± 0.18 [a] | 2.21 ± 0.10 [c] |
| | Thymol | 50 | 90.2 ± 1.4 [a] | 9.07 ± 0.36 [a] | 3.36 ± 0.16 [a] |
| | | 100 | 89.6 ± 2.3 [a] | 9.04 ± 0.22 [a] | 3.32 ± 0.12 [a] |
| | | 200 | 82.7 ± 1.1 [ab] | 8.95 ± 0.17 [a] | 3.18 ± 0.15 [a] |
| | *p*-Cymene | 50 | 91.4 ± 2.7 [a] | 9.09 ± 0.20 [a] | 3.37 ± 0.15 [a] |
| | | 100 | 91.3 ± 1.4 [a] | 9.05 ± 0.22 [a] | 3.37 ± 0.09 [a] |
| | | 200 | 89.8 ± 1.8 [a] | 9.04 ± 0.16 [a] | 3.36 ± 0.15 [a] |
| *J. procera* | EO | 50 | 91.4 ± 1.4 [a] | 9.02 ± 0.38 [a] | 3.26 ± 0.16 [a] |
| | | 100 | 90.8 ± 1.9 [a] | 8.96 ± 0.24 [a] | 3.13 ± 0.12 [a] |
| | | 200 | 88.3 ± 2.1 [a] | 8.63 ± 0.19 [a] | 2.69 ± 0.11 [ab] |
| | Eugenol | 50 | 91.9 ± 2.9 [a] | 9.03 ± 0.38 [a] | 3.30 ± 0.16 [a] |
| | | 100 | 89.6 ± 1.6 [a] | 8.99 ± 0.24 [a] | 3.25 ± 0.12 [a] |
| | | 200 | 84.1 ± 2.3 [ab] | 8.94 ± 0.19 [a] | 3.08 ± 0.11 [a] |
| | *β*-Caryophyllene | 50 | 91.6 ± 1.5 [a] | 9.08 ± 0.18 [a] | 3.31 ± 0.15 [a] |
| | | 100 | 91.1 ± 1.6 [a] | 9.07 ± 0.00 [a] | 3.24 ± 0.12 [a] |
| | | 200 | 90.4 ± 1.4 [a] | 9.04 ± 0.33 [a] | 3.11 ± 0.11 [a] |
| | *α*-Pinene | 50 | 91.3 ± 1.4 [a] | 9.09 ± 0.18 [a] | 3.33 ± 0.15 [a] |
| | | 100 | 91.1 ± 1.6 [a] | 9.07 ± 0.00 [a] | 3.21 ± 0.12 [a] |
| | | 200 | 90.7 ± 1.6 [a] | 9.05 ± 0.33 [a] | 3.06 ± 0.11 [a] |
| | Control | | 91.3 ± 1.4 [a] | 9.08 ± 0.28 [a] | 3.37 ± 0.15 [a] |
| | *F* value | - | 8.2 | 2.4 | 2.2 |
| | *P* value | - | 0.0072 | 0.0034 | 0.008 |

* Values are the mean ± S.E. of four replicates; RL = radicle growth (length of seeds, cm); SL = shoot length (cm). In the same column, means followed by the same letters are not significantly different ($p \leq 0.05$) (Tukey's HSD test). All *F*-values are significant at $p \leq 0.001$.

## 4. Discussion

The aroma profiles of the EOs showed substantial similarities to those previously reported. For example, eugenol was the key monoterpene in *J. procera* EO in earlier reports [38,39]. Our results regarding the EO profile of *T. vulgaris* match well with those of earlier reports [40,41]. Differences in the yield and/or main terpenes of the tested EOs and the same species growing in different geographical regions were recorded [42–45]. These variations are attributed to environmental (climatical, seasonal, and geographical) and genetic factors. The harvesting time, extracted part, and extraction procedures are also major issues [2,3,42].

The bioactivity of EOs against insect pests is an important property that enables the use of phytochemicals as bio-rational and eco-friendly pesticides to control harmful insects in storage ecosystems in order to reduce the extensive use of pesticides. Our results clarify that *T. vulgaris* and *J. procera* EOs and bioactive terpenes have remarkable pest control potential against the khapra beetle. Different EOs were assessed as grain protectants against harmful insects, including *T. granarium*. However, this is considered the first report of the bioactivities of the EOs of Saudi *T. vulgaris* and *J. procera* against *T. granarium*. Our results agreed with those of many authors who tested the bioactivities of EOs from other plants against *T. granarium*, such as *Moringa oleifera*, *Datura stramonium*, *Eucalyptus camaldulensis*, and *Nigella sativa* [46], and *Achillea santolina*, *A. biebersteinii* and *A. mellifolium* [2]. According

to our results, the adults were more susceptible to the EOs than the larvae. Similar results were recorded by many authors. *T. granarium* was susceptible to EOs via contact and fumigation bioactivities. The entry mechanism of toxins is a detrimental factor that affects bioactivity against pests [2,3]. For fumigants, insects in mobile stages (adults and non-diapausing larvae) are more sensitive than those in sedentary stages (eggs and pupae), due to variations in respiratory rate [2]. These findings match well with our results, where adults exhibited a great sensitivity to botanicals.

Herein, the monoterpenes in the EOs were potent insecticides. As previously established, the bioactivities of the EOs against insects might be correlated with their contents of monoterpenes with proven insecticidal bioactivities, such as eugenol, carvacrol, thymol, *p*-cymene, $\gamma$-terpinene, 1,8 cineol, terpinene-4-ol and carvone [6,10,26,47,48]. Among the monoterpenes in the thyme EO studied in the present study, carvacrol and thymol showed pronounced bioactivities against *T. granarium*. These monoterpenes have received special attention in EO research, and they have been widely evaluated as pesticides against many pests, including those that infest stored grains [13,16,47–53]. The insecticidal properties of carvacrol and thymol correlated with their abilities to cause disorganization in bio-membranes, leading to the loss of permeability [49]. However, carvacrol presented a significantly higher insecticidal effect than thymol. These results match well with those of earlier reports [53,54]. The only structural difference between these two monoterpenes is the ortho-position of the OH group on the benzene ring of the larger aliphatic chain, and this may be related to their difference in activity [54]. The hydrophobic nature of thymol and carvacrol enables them to interact with the lipid bilayer of bio-membranes, causing the loss of membrane integrity, and the leakage of cellular materials, such as ions, nucleic acids, and ATP, enhancing their bioactivities [55]. Our results on the insecticidal bioactivity of eugenol, the key monoterpenoid in *J. procera* EO, match well with those of earlier reports, where this monoterpene exhibited insecticidal bioactivity against different pests, including stored grain pests [13,55,56].

The tested EOs had comparatively stronger bioactivities than any of their bioactive components. Recent investigations have clarified that the activities of EOs might be influenced by the interactions among different components of the EO, where minor fractions possess a critical role, due to an additive and/or synergistic interaction among the EO compartments [2,3,6,10,18,57], in which several EO components participate in cell penetration, fixation on cell membranes and distribution in biosystems. As is known, the major terpene in an EO mixture is preferentially metabolized and detoxified by the insect, while the minority component is poorly detoxified by the $P_{450}$ biosystem [6]. Hence, a synergistic interaction among minor and major compartments in a plant oil commonly occurs.

The EOs and terpenes significantly inhibited larval AChE activity. In the literature, the insecticidal bioactivities of EOs and their components, particularly monoterpenes, are attributed to their ability to inhibit AChE activity, blocking the insect's octopamine receptors or GABA-gated chloride channels, and binding to the insect's midgut receptors, causing cell expansion, rupture, and ion leakage through holes or ion channels in the gut membrane, and they can also cause the leakage of basic cations (such as potassium ions) from the host mitochondria [6,10,17,58]. Therefore, the tested oils may exert their activity via one or more of these mechanisms of action.

The EO products were safe for the earthworm, *E. fetida*, even at high concentrations, and also had neglectable impacts on wheat grain viability. Several studies have provided evidence that phytochemicals and plant-based products are relatively safe for non-target organisms, when tested as bio-rational pesticides against harmful insects [23,59–61]. However, most studies usually focus on acute toxicity evaluations, and neglect sub-chronic and chronic assessments.

According to Sigma-Aldrich [62], the rat oral $LD_{50}$ values of carvacrol, thymol, and *T. vulgaris* EO are 810, 980 and 2840 mg/kg, respectively. Additionally, the dermal rat $LD_{50}$ of *T. vulgaris* EO and thymol exceeds 2000 mg/kg, while reaching 5000 mg/kg against rabbits. Thus, thyme oil and its main monoterpenes have low acute mammalian toxicity.

Interestingly, thymol and carvacrol are approved by the FDA (Food and Drug Agency, Silver Spring, MD, USA), and they are listed as safe chemical flavorings, together with the thyme plant.

## 5. Conclusions

Overall, the results of the present study indicate that *T. vulgaris* and *J. procera* essential oils, as well as their major terpenes, demonstrate significant insecticidal and acetylcholinesterase inhibitory activities against the khapra beetle, one of the most serious and destructive coleopteran insects of stored grains. The tested oils and terpenes were safe for the earthworm, *E. fetida*, and had no effect on wheat grain viability. The fumigant action of the test materials and their biosafety potentiate their possible use for controlling harmful insects in grain stores. However, further investigations are needed to fully establish the biosafety of the tested botanicals before their incorporation in insect control protocols.

**Author Contributions:** Conceptualization, G.N.; methodology, validation and investigation, G.N. and A.A.; formal analysis, G.N.; resources, A.A.; data curation, G.N. and A.A.; writing—original draft preparation, G.N.; writing—review and editing, G.N. and A.A.; visualization G.N.; supervision, G.N.; project administration, G.N.; funding acquisition, A.A. All authors have read and agreed to the published version of the manuscript.

**Funding:** This work has been funded by the Deputy for Research and Innovation, Ministry of Education, Kingdom of Saudi Arabia (grant no. NU/IFC/ENT/01/003) under the Committee of Institutional Funding at Najran University.

**Data Availability Statement:** The datasets of this study are available upon request.

**Conflicts of Interest:** The authors declare no known conflict of interest.

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
