# Peer review of "Chemical Profile, Bioactivity, and Biosafety Evaluations of Essential Oils and Main Terpenes of Two Plant Species against Trogoderma granarium"

_agronomy, doi:10.3390/agronomy12123112_

Round 1
Reviewer 1 Report
This article evaluated the effect of essential oils against pests and earthworm, the authors also examined biosafety of these essential oils and determined that the used essential oils exhibit potential against pests and can be used as pesticide. Before recommending this article for publication, there are some shortcomings for that should be resolve.
Abstract
Line 13 why GC-MS is written in the bracket.
Line 18-23 should be revise these sentences contain grammatical errors.
Replace potentiality with potential line 24.
Methods are not properly presented in the abstract.
Add future recommendations at the end of the abstract.
Introduction
Line 37 write proper taxonomy of the Khapra beetle.
The author covered only one aspect of the title in the introduction.
There is very little information about essential oils, and the plants selected for extraction of essential oils by citing relevant studies. Some are recommended here. https://doi.org/10.1016/j.pmpp.2021.101639, http://doi.org/10.36899/JAPS.2022.3.0484,
The authors should present significance of EOs specifically in terms of pesticides, also add taxonomy, distribution and essential oils potential of the Juniperus procera and Thymus vulgaris.
Line 58-59 add mechanism of action of insecticide.
Provide reasons How EOs are better than synthetic pesticides and insecticides.
Aims and objective line 59-63 must be revised and clarify.
Methods
Line 160 italicized the sp. name.
Results and discussion
Results are well presented but discussion lack comparison with other studies.
Conclusion
Conclusions must be based on results not only recommendations.
Author Response
Dear Dr.
I would like to thank you and the reviewers for all the valuable comments and constructive suggestions on the Manuscript: ID: agronomy-2047138, titled "Chemical Profile, Bioactivity and Biosafety Evaluations of Essential Oils and Main Terpenes of Two Plant Species against Trogoderma granarium". In this revised form of the manuscript (R1), I considered all comments of the editor and reviewers. Please find each of these comments in conjugation with my response (point by point). For consistency, I preferred to make the corrections in the word file, and all of the corrected words and/or statements are highlighted in a red color in the revised manuscript.
Please find attached the revised version of the manuscript, which I would like to submit for your kind consideration.
-------------------------------------------------------------------------------------------------------
Reviewer #1:
Abstract
Comment: Line 13 why GC-MS is written in the bracket.
Response: corrected.
Comment: Line 18-23 should be revise these sentences contain grammatical errors.
Response: Done. The sentences have been re-phrased
Comment: Replace potentiality with potential line 24.
Response: Done
Comment: Methods are not properly presented in the abstract.
Response: Corrected
Comment: Add future recommendations at the end of the abstract.
Response: Done
Introduction
Comment: Line 37 write proper taxonomy of the Khapra beetle.
Response: Done
Comment: The author covered only one aspect of the title in the introduction.
Response: This comment is taken into consideration. In this revised form of the manuscript we made a highlight on the advantages of plant oils as natural pest control tools in addition to other related aspects.
Comment: There is very little information about essential oils, and the plants selected for extraction of essential oils by citing relevant studies. Some are recommended here. https://doi.org/10.1016/j.pmpp.2021.101639, http://doi.org/10.36899/JAPS.2022.3.0484,
Response: Information on essential oils, and the tested plants are addressed, and the suggested articles are cited.
Comment: The authors should present significance of EOs specifically in terms of pesticides, also add taxonomy, distribution and essential oils potential of the Juniperus procera and Thymus vulgaris.
Response: All these topics are well-covered in this revised version of the manuscript (Done).
Comment: Line 58-59 add mechanism of action of insecticide.
Response: Done
Comment: Provide reasons How EOs are better than synthetic pesticides and insecticides.
Response: This topic is well-adressed in this revised version of the manuscript (Done).
Comment: Aims and objective line 59-63 must be revised and clarify.
Response: The objectives of the study are re-written and clarified. (Done).
Methods
Comment: Line 160 italicized the sp. name.
Response: Done
Results and discussion
Comment: Results are well presented but discussion lack comparison with other studies.
Response: Done
Conclusion
Comment: Conclusions must be based on results not only recommendations.
Response: The conclusion is re-phrased (Done).
Finally, I am greatly appreciated for thr reviewer for the valuable and constructive comments on the manuscript.
Reviewer 2 Report
These are my main comments on the manuscript (agronomy-2047138) entitled “Chemical Profile, Bioactivity and Biosafety Evaluations of Essential Oils and Main Terpenes of Two Plant Species against Trogoderma granarium”. The manuscript investigates the pesticidal and AChE inhibition bioactivities of Juniperous procera and Thymus vulgaris and their bioactive terpenes against T. granarium. Following substantial revisions should be incorporated in the manuscript prior to acceptance.
1. I have concerns about the manuscript sections that I believe need to be addressed in order to improve its clarity.
2. A hypothesis for this work is needed.
3. Conclusion section should be rephrased.
4. Other revisions could be checked in PDF attached.

Author Response
Dear Dr.
I would like to thank you and the reviewers for all the valuable comments and constructive suggestions on the Manuscript: ID: agronomy-2047138, titled "Chemical Profile, Bioactivity and Biosafety Evaluations of Essential Oils and Main Terpenes of Two Plant Species against Trogoderma granarium". In this revised form of the manuscript (R1), I considered all comments of the editor and reviewers. Please find each of these comments in conjugation with my response (point by point). For consistency, I preferred to make the corrections in the word file, and all of the corrected words and/or statements are highlighted in a red color in the revised manuscript.
Please find attached the revised version of the manuscript, which I would like to submit for your kind consideration.
Reviewer #2:
Comment: These are my main comments on the manuscript (agronomy-2047138) entitled “Chemical Profile, Bioactivity and Biosafety Evaluations of Essential Oils and Main Terpenes of Two Plant Species against Trogoderma granarium”. The manuscript investigates the pesticidal and AChE inhibition bioactivities of Juniperous procera and Thymus vulgaris and their bioactive terpenes against T. granarium. Following substantial revisions should be incorporated in the manuscript prior to acceptance.
- I have concerns about the manuscript sections that I believe need to be addressed in order to improve its clarity.
- A hypothesis for this work is needed.
- Conclusion section should be rephrased.
- Other revisions could be checked in PDF attached.
Response: I am greatly appreciated for the reviewer for your effort and constructive opinion. In this revised form of the manuscript, I considered all of your comments as you are kindly attached in the pdf file, which really improve the manuscript.
Reviewer 3 Report
1. Line 69 - proofreading 0C
2. Khapra beetle was reared in pesticides-free environment for several generations using sterilized wheat grains (12% moisture) as media, how several generations?
3. Aroma Profile of EOs
4. Eugenol
Yellowish oil. 1H NMR (600 MHz, CDCl3): δ 6.82 (H-6, d, J = 8.4 Hz), 6.65-6.69 (2H, m, H3, H-5), 3.31 (H2-1′, d, J = 6.9 Hz), 5.92 (H-2′, m), 5.04-5.06 (H2-3′, m), 5.46 (s, 1-OH), 3.84 (2-OCH3, s); 13C NMR (150 MHz, CDCl3): δ 142.86 (C-1), 147.30 (C-2), 111.09 (C-3), 131.81 (C-4), 121.19 (C-5), 113.90 (C-6), 38.98 (C-1′), 136.88 (C-2′), 115.41 (C-3′), 55.74 (2-OCH3) (28del Fierro et al., 2012)
êžµ-Caryophyllene
Colorless oil. 1H NMR (600 MHz, CDCl3) δ: 1.67 (H2-1), 1.43, 1.53 (H2-2), 1.91, 2.08 (H2-3), 5.32 (H-5, dd, 4.1, 9.7 Hz), 2.02, 2.37 (H2-6), 2.03, 2.24 (H2-7), 2.31 (H-9); 1.61, 1.67 (H2-10), 0.97 (H3-12), 0.95 (H3-13), 1.61 (H3-14), 4.83, 4.94 (H2-15); 13C NMR (150 MHz, CDCl3) δ: 53.6 (C-1), 29.4 (C-2), 39.8 (C-3), 135.4 (C-4), 123.8 (C-5), 28.7 (C-6), 34.6 (C-7), 154.8 (C-8), 48.4 (C-9), 40.2 (C-10), 33.1 (C-11), 22.6 (C-12), 30.2 (C-13), 16.2 (C-14), 111.7 (C-15). (29Ragasa et al. 2013).
α-pinene
Colorless oil. 1H NMR (CDCl3 , 300 MHz): δ 1.92 (m, 1H), δ 1.4 (s, 3H), δ 1.6 (s, 3H), δ 1.8 (s, 3H), δ 1.9 (m, 2H), δ 2.3 (m, 1H), δ 2.4 (m, 1H), δ 4.2 (s, 1H), δ 5.6 (t, 1H), 20.67, 22.35, 22.47, 26.36, 32.62, 36.08, 68.03, 94.15, 123.87, 134.38; 13C NMR (125 MHz, CHCl3):δ 46.99(C-1), 144.6 (C-2), 116.0 (C-3), δ 31.3 (C-4), 40.69(C-5), 37.97(C-6),31.5 (C-7), δ 26.3 (C-8), 20.8 (C-9), 23.01 (C-10) (30Lee, 2002; 31Matsuo et al., 2011). I suggest the presentation in another form.
Author Response
Dear Dr.
I would like to thank you and the reviewers for all the valuable comments and constructive suggestions on the Manuscript: ID: agronomy-2047138, titled "Chemical Profile, Bioactivity and Biosafety Evaluations of Essential Oils and Main Terpenes of Two Plant Species against Trogoderma granarium". In this revised form of the manuscript (R1), I considered all comments of the editor and reviewers. Please find each of these comments in conjugation with my response (point by point). For consistency, I preferred to make the corrections in the word file, and all of the corrected words and/or statements are highlighted in a red color in the revised manuscript.
Please find attached the revised version of the manuscript, which I would like to submit for your kind consideration.
Reviewer 3
Line 69 - proofreading 0C
Khapra beetle was reared in pesticides-free environment for several generations using sterilized wheat grains (12% moisture) as media, how several generations?
Eugenol
Yellowish oil. 1H NMR (600 MHz, CDCl3): δ 6.82 (H-6, d, J = 8.4 Hz), 6.65-6.69 (2H, m, H3, H-5), 3.31 (H2-1′, d, J = 6.9 Hz), 5.92 (H-2′, m), 5.04-5.06 (H2-3′, m), 5.46 (s, 1-OH), 3.84 (2-OCH3, s); 13C NMR (150 MHz, CDCl3): δ 142.86 (C-1), 147.30 (C-2), 111.09 (C-3), 131.81 (C-4), 121.19 (C-5), 113.90 (C-6), 38.98 (C-1′), 136.88 (C-2′), 115.41 (C-3′), 55.74 (2-OCH3) (28del Fierro et al., 2012)
êžµ-Caryophyllene
Colorless oil. 1H NMR (600 MHz, CDCl3) δ: 1.67 (H2-1), 1.43, 1.53 (H2-2), 1.91, 2.08 (H2-3), 5.32 (H-5, dd, 4.1, 9.7 Hz), 2.02, 2.37 (H2-6), 2.03, 2.24 (H2-7), 2.31 (H-9); 1.61, 1.67 (H2-10), 0.97 (H3-12), 0.95 (H3-13), 1.61 (H3-14), 4.83, 4.94 (H2-15); 13C NMR (150 MHz, CDCl3) δ: 53.6 (C-1), 29.4 (C-2), 39.8 (C-3), 135.4 (C-4), 123.8 (C-5), 28.7 (C-6), 34.6 (C-7), 154.8 (C-8), 48.4 (C-9), 40.2 (C-10), 33.1 (C-11), 22.6 (C-12), 30.2 (C-13), 16.2 (C-14), 111.7 (C-15). (29Ragasa et al. 2013).
α-pinene
Colorless oil. 1H NMR (CDCl3 , 300 MHz): δ 1.92 (m, 1H), δ 1.4 (s, 3H), δ 1.6 (s, 3H), δ 1.8 (s, 3H), δ 1.9 (m, 2H), δ 2.3 (m, 1H), δ 2.4 (m, 1H), δ 4.2 (s, 1H), δ 5.6 (t, 1H), 20.67, 22.35, 22.47, 26.36, 32.62, 36.08, 68.03, 94.15, 123.87, 134.38; 13C NMR (125 MHz, CHCl3):δ 46.99(C-1), 144.6 (C-2), 116.0 (C-3), δ 31.3 (C-4), 40.69(C-5), 37.97(C-6),31.5 (C-7), δ 26.3 (C-8), 20.8 (C-9), 23.01 (C-10) (30Lee, 2002; 31Matsuo et al., 2011). I suggest the presentation in another form.
Response: I am greatly appreciated for the reviewer for the constructive opinion. In this revised form of the manuscript, I considered all of your comments. Thank you again
Round 2
Reviewer 1 Report
I have no further comments.
Author Response
Comment: have no further comments.
Response: I would like to thank the reviewer for the constructive opinion. Thank you again.
Reviewer 2 Report
The authors have incorporated all suggestions and comments into the revised version, now the manuscript seems much clear. There is minor point to be corrected:
L.13: Define “GC–MS”
Ls.27-28: Keywords should be in alphabetic order. Also, keywords serve to widen the opportunity to be retrieved from a database. To put words that already are into title and abstracts makes KW not useful. Please choose terms that are neither in the title nor in abstract.
Ls.71 and 75: Juniperous procera and Thymus vulgaris should be J. procera and T. vulgaris
L.106: …equal to 1 mL/min…
L.160: … papers, 7 cm in diameter
L.210: Delete “.” before 3
L.359: Tukey's HSD test
L.389: Delete “the bioactivity.”
L.407: The insecticidal properties of carvacrol…
Author Response
Dear Dr.
I would like to thank you and the reviewers for all the valuable comments and constructive suggestions on the Manuscript: ID: agronomy-2047138, titled "Chemical Profile, Bioactivity and Biosafety Evaluations of Essential Oils and Main Terpenes of Two Plant Species against Trogoderma granarium". In this revised form of the manuscript (R2), I considered all comments of the editor and reviewers. Please find each of these comments in conjugation with my response (point by point). For consistency, I preferred to make the corrections in the word file, and all of the corrected words and/or statements are highlighted in a red color in the revised manuscript.
Please find attached the revised version of the manuscript, which I would like to submit for your kind consideration.
---------------------------------------------------------------------------------------
Reviewer #2:
Comment: The authors have incorporated all suggestions and comments into the revised version, now the manuscript seems much clear. There is minor point to be corrected.
Response: I would like to thank the reviewer for the constructive opinion. In this revised form of the manuscript, I considered all of your comments. Thank you again
Comment: L.13: Define “GC–MS”
Response: Done
Comment: Ls.27-28: Keywords should be in alphabetic order. Also, keywords serve to widen the opportunity to be retrieved from a database. To put words that already are into title and abstracts makes KW not useful. Please choose terms that are neither in the title nor in abstract.
Response: Done
Comment: Ls.71 and 75: Juniperous procera and Thymus vulgaris should be J. procera and T. vulgaris
Response: Done
Comment: L.106: …equal to 1 mL/min…
Response: Done
Comment: L.160: … papers, 7 cm in diameter
Response: Done
Comment: L.210: Delete “.” before 3
Response: Done
Comment: L.359: Tukey's HSD test
Response: Done
Comment: L.389: Delete “the bioactivity.”
Response: Done
Comment: L.407: The insecticidal properties of carvacrol…
Response: Done